# A Reinforcement Learning Based
# Universal Sequence Design for Polar Codes

## Abstract

To advance Polar code design for 6G applications, we develop a reinforcement learning–based universal sequence design framework that is extensible and adaptable to diverse channel conditions and decoding strategies. Crucially, our method scales to code lengths up to 2048, making it suitable for use in standardization. Across all $(N, K)$ configurations supported in 5G, our approach achieves competitive performance relative to the NR sequence adopted in 5G and yields up to a 0.2 dB gain over the beta-expansion baseline at $N = 2048$. We further highlight the key elements that enabled learning at scale: (i) incorporation of physical law constrained learning grounded in the universal partial order property of Polar codes, (ii) exploitation of the weak long term influence of decisions to limit lookahead evaluation, and (iii) joint multi-configuration optimization to increase learning efficiency.

## 1. Introduction

### 1.1. Polar Codes in 5G NR and beyond

Polar codes are an essential component of the 5G NR standard, responsible for ensuring the integrity of uplink control information (UCI) transmitted over the physical uplink control channel (PUCCH) and the physical uplink shared channel (PUSCH) in the uplink. Similarly in the downlink, Polar codes protect the integrity of downlink control information (DCI) transmitted over the physical downlink control channel (PDCCH), as well as the master information block (MIB) transmitted over the physical broadcast channel (PBCH).

As the industry moves towards 6G, there is an anticipated demand for larger DCI payloads to support additional features,

increased channel state information feedback and other signaling needs. In addition, stronger error protection will be necessary to enhance cell-edge user experience. The Polar codes and the associated universal sequence defined in 5G NR (Bioglio et al., 2020) are currently limited to a block length of 1024 bits. This restricts the maximum payload size and the number of parity bits available for improved protection under poor channel conditions. For large payloads, the current 5G NR strategy segments the data across multiple, smaller code blocks; however, these smaller codes cannot exploit the additional polarization layers available at larger block lengths. For enhanced protection, the standard resorts to suboptimal repetition coding. Extending Polar code lengths can unlock performance gains unattainable in the current specification.

For a Polar code to function optimally under successive cancellation list (SCL) decoding, the information bits must be assigned to the synthetic channels in a way that minimizes the overall error probability. As an example, consider a $(128, 36)$ Polar code with code length $N = 128$ bits and payload size $K = 36$ bits. Of the 128 synthetic channels, one must define a mapping to optimally assign the 36 payload bits and freeze the remaining bit channels by setting the input to zero. The resultant mapping, referred to as a frozen bitmap, must be specified for each $(N, K)$ configuration supported by the standard. Given the large variability of channel conditions and physical resource constraints in a multi-user wireless environment, a wide range of code lengths and all possible code rates (defined by $CR = K/N$) must be supported. The 5G NR standard supports $N \in \{32, 64, 128, 256, 512, 1024\}$ and $K = 12$ and up to the number of physically transmitted bits.

Polar codes share the same Kronecker product construction as Reed-Muller (RM) codes (Reed, 1954; Muller, 1954); however, their bit channel selection mechanism and optimizing objective are markedly different. In RM codes, the rows of the generator matrix are selected based on the largest Hamming weights, yielding a channel independent design that maximizes minimum distance, which is a desirable property for short codes. In contrast, Polar codes select rows by maximizing conditional mutual information (or equivalently, minimizing the Bhattacharyya parameter), re-

---

[1]Anonymous Institution, Anonymous City, Anonymous Region, Anonymous Country. Correspondence to: Anonymous Author <anon.email@domain.com>.

Preliminary work. Under review by the International Conference on Machine Learning (ICML). Do not distribute.

sulting in a channel-specific design that minimizes the error probability under successive cancellation (SC) decoding. Invented by Arıkan in 2009 (Arikan, 2009), Polar codes is the first channel code with an explicit construction that provably achieves the channel capacity. In practice, Polar codes offers superior performance with low-complexity decoding algorithms, making them indispensable for 5G control channels where they outperform LDPC codes at short block lengths. We refer interested readers to this excellent introduction to Polar codes.[1]

### 1.2. Adoption of universal reliability sequence

The sheer number of possible bit mappings presents a significant challenge for storage requirements and computational burden if they are generated on demand. However, if the problem can be simplified such that each synthetic channel is associated with a reliability metric and these channels be ranked universally by this metric for all block lengths, then the problem reduces to designing a single universally applicable sequence for all $(N, K)$ combinations. For each $(N, K)$, the assignment method reduces to finding all channel indices less than N and selecting out of this set the K most reliable indices to carry the information bits. Only a single maximum length N sequence is stored in memory, substantially reducing both storage and computational complexity. This universal sequence approach has been adopted in the 5G NR standard (Bioglio et al., 2020).

### 1.3. Reinforcement learning in code construction

Noting that existing code construction methods do not produce optimal codes for Polar SCL decoding, (Liao et al., 2021) began the exploration of using reinforcement learning to learn the bit assignment directly from the outcome of a Polar SCL decoder. This early approach used a Tabular RL method (SARSA) to learn the frozen bit assignment for a fixed payload size, a fixed code length and a fixed signal-to-noise ratio (SNR). Because tabular RL lacks generalization capability across code configurations, this method is not scalable to a large number of codes. In mobile communication systems, the dynamic and wide ranging fading environments requires operations over diverse code rates, code lengths and SNR regimes. Similar attempts by (Mishra et al., 2022) approached the PAC code construction problem using a modified Q-learning algorithm, which inherits the same limitations of Tabular RL. Moreover, their use of the Reed-Muller initialization rule severely limits the algorithm's ability to learn optimal assignments. As shown experimentally, the RM rule becomes highly suboptimal at moderate code lengths. To address the scalability limitations of Tabular methods, (Liao et al., 2023) applied a deep-RL approach based on Deep Q-learning with a graph neural

network (GNN) approximating the action-value function. The algorithm utilized a heterogeneous-GNN, where predicted values are refined through multiple stages of message passing among graph nodes. Our approach was inspired by this powerful deep reinforcement learning technique and the promise of scalability of this work, although we note that the training complexity of the proposed method limited its use to short code lengths.

### 1.4. Universal Partial Order

Building upon Arıkan's foundational work, researchers discovered the universal partial order (UPO), a set of relations that governs the reliability among the synthetic bit channels and remains valid irrespective of the underlying physical channel characteristics under the assumption of SC decoding (Schürch, 2016). (He et al., 2017) exploited the UPO relations to develop an efficient sequence construction scheme based on beta-expansion theory. Subsequent studies (Yao et al., 2024; Liu et al., 2024) extended this line of research by discovering new partial order relations that enable improved performance for sequentially rate-matched Polar codes.

### 1.5. Physical law constrained machine learning

Many problems in material sciences and physics concern the learning of phenomena and the generation of entities that adhere to known geometric structure or physical laws. While an ML agent may learn the distribution of the training data with a high degree of accuracy, it might struggle to generate novel samples that exhibit specific desired properties, particularly when such samples are underrepresented or absent in the dataset. However, because of the enormous space involved, a guided strategy is needed to constrain the agent to generate samples within the distribution of interest. This motivated (Okabe et al., 2025) to incorporate geometric constraints in the SCIGEN model, thereby improving the quality of generated samples. In physics, even with abundant observational data, purely data-driven models often produce predictions that are physically inconsistent or implausible. Therefore, it is essential that fundamental physical laws and domain knowledge be incorporated in the model as an informative prior. Physics-informed learning (Karniadakis et al., 2021) advocates the inclusion of physical laws based constraints either as inductive bias (implemented as hard constraints) or as learning bias (implemented as penalty terms). This approach enhances generalization in the small data regime by restricting deep learning models to operate in lower-dimensional manifolds consistent with theory. Guided by this principle, we leverage the known UPO relations that govern the relative reliability of the synthetic channels to improve the predictive performance of the RL agent while keeping the data demand modest.

---

[1]Polar codes tutorial from Prof. Balatsoukas-Stimming

### 1.6. Contributions

We present the first openly-documented RL-based universal sequence design method that is scalable to any code lengths, trainable within reasonable time budgets using standard GPU/CPU compute infrastructure, and adaptable to any channel conditions and decoding strategies. Our approach integrates a deep reinforcement learning method, specifically Proximal Policy Optimization (PPO) (Schulman et al., 2017) to eliminate the need for an experience replay buffer and improve training stability, constrains the search space using the UPO relations to make the optimization task tractable, embeds lower-N sequences to preserve the optimality of performance across smaller block lengths, adopts a Monte-Carlo-Tree-Search (MCTS) inspired iterative learning strategy to limit lookahead evaluation, and applies progressive constraint relaxation to improve the quality of the result. Moreover, we introduce a joint optimization framework that promotes knowledge transfer among different configurations, significantly improving learning efficiency. Our method achieves competitive performance across all $(N, K)$ configurations supported in 5G NR relative to the NR sequence and delivers up to 0.2 dB gain over beta-expansion at $N = 2048$. Source code will be made available at the time of publication.

## 2. Problem Definition

Given a propagation channel environment and a Polar decoding algorithm, with limits $N_{max}, N_{min}$ and $K_{min}$, we seek to find an optimal stochastic policy where the sampled instances produce, with high probability, an absolute ordering (the universal sequence) that outperforms the beta-expansion algorithm (baseline) in terms of block error rate (BLER)[2] over all permissible block length, payload combinations, i.e. for all $(N, K)$ Polar codes derived using the absolute ordering, where $K_{min} \leq K < N$, $N_{min} \leq N \leq N_{max}$ ($N$ must be an integer power of 2).

A subtle but important aspect of the problem is that bit indices participate in the frozen bitmaps of codes from multiple block lengths. Thus effectively we have a multi-objective optimization problem without a unique dominant winner. To resolve this ambiguity, we explicitly prioritize the performance at smaller block lengths over larger ones.

In addition to achieving the main objective, the proposed method must be computationally scalable to practical block lengths (up to $N_{max} = 2048$), trainable within a reasonable time budget. Scalability remains a major challenge for existing RL-based approaches.

---

[2]In practice the relative BLER performance is measured as the SNR gap between two schemes at some fixed BLER e.g. 0.01.

## 3. Proposed Method

### 3.1. High level description

We first provide a high-level overview of the construction method before detailing each step. The sequence is constructed in a nested, iterative manner, starting from $N = N_{min}$. The sequence generated at each stage serves as a sub-sequence that is embedded within the next length progression $N_{next} = 2N$. This ensures that the performance at the current N is preserved. At each step, an RL-based neural-assisted search is conducted to identify a candidate sequence that jointly maximizes the performance of all length-N polar codes. It is important to note that due of the universal reliability ranking, for any fixed $N$, the first $J$ bit channel decisions influence the performance of $(N, K)$ Polar codes for all $K > J$. The UPO property of Polar codes is exploited to constrain the search space, keeping the optimization computationally feasible. Two additional relaxation techniques are incorporated to expand the search space selectively, allowing the discovery of higher-performing candidates.

### 3.2. Action space constraint management

To scale the proposed approach to $N_{max} = 2048$ (denoted N2048), it is critical to adopt a principled method of restricting the search space, without which the search becomes computationally intractable. Using UPO, we reduced the state space for N2048 from $10^{5894}$ to $10^{2582}$. In addition, without any knowledge of the previous lower-N result, the current $N$ search will produce a suboptimal sequence at lower-N. Thus, it is essential that we enforce the ordering of indices in the lower-N search, which preserves the performance at lower-N. We refer to this technique as *lower-N embedding*. However, we found empirically that lower-N embedding leads to degradation in performance at large block-lengths. From in-depth analysis at $N$, the preferred ordering of the initial portion of the lower-N indices $i \in \{j \mid j < N_{lower}\}$ is observed to deviate from that of the lower-N sequence. Because the standard defines a minimum payload size for UCI/DCI, we exploit this by relaxing the constraint of the first $K_{min}$ imposed by the lower-N sequence, since the relative ordering of these bits do not influence the performance at $N_{lower}$. Thus we introduce this as the first relaxation step. Second, since UPO is derived under the assumption of SC decoding rather than SCL decoding, relaxing the UPO constraint can expose high-performing candidates under SCL decoding. In addition to the immediate neighbors in the UPO lattice, we extend the action space to include one-hop neighbor nodes that would normally violate UPO consistency through the process of *node promotion*. We refer to the combined constraints as the *UPO+ rule*. Figure 1 illustrates this concept.

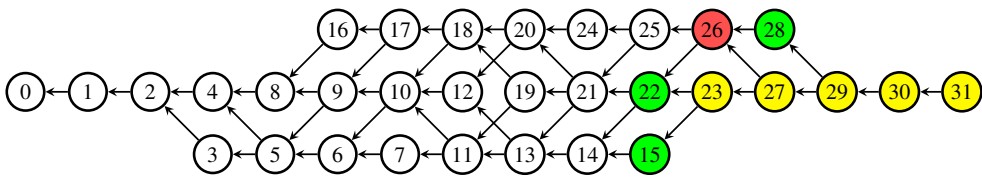

*Figure 1.* UPO lattice with the current context in yellow, the immediate neighborhood in green and the disallowed neighbor in red. Our proposed relaxation will include the red node as permissible action in the current context, as this is the immediate neighbor of node 27.

### 3.3. Universal partial order

We restate the universal partial order result here for convenience.

**Definition 3.1.** A synthetic channel $i$ is more reliable than $j$ as measured by the mutual information or equivalently the Bhattacharyya parameter (Arikan, 2009), is denoted $i \succ j$.

Given any pair of synthetic channel indices $(x, y)$, their reliability relation is determined by the following rules (Mondelli et al., 2018):

- **Addition**: If a binary representation of the index of a synthetic channel is $(a, b, 0, c)$, then it must be less reliable than the synthetic channel whose index has the binary representation $(a, b, 1, c)$. This relation holds for one or more consecutive bits. For example:

$$4\ (1, 0, \underline{0}) \quad \prec \quad 5\ (1, 0, \underline{1}),$$
$$9\ (1, \underline{0, 0}, 1) \quad \prec \quad 15\ (1, \underline{1, 1}, 1).$$

- **Left-swap**: If a binary representation of the index of a synthetic channel is $(a, 0, b, 1, c)$, then it must be less reliable than the synthetic channel whose index has a binary representation $(a, 1, b, 0, c)$. This pattern can appear multiple times, and the $0, 1$ pair do not need to be adjacent to each other. For example:

$$3\ (\underline{0}, \underline{1}, 1) \quad \prec \quad 5\ (\underline{1}, \underline{0}, 1),$$
$$13\ (\underline{0}, 1, \underline{1}, 0, 1) \quad \prec \quad 25\ (\underline{1}, 1, \underline{0}, 0, 1).$$

The nested and symmetric property of UPO leads to an efficient recursive construction of these relations, as demonstrated in section II-B of (He et al., 2017). This structure forms the theoretical foundation upon which our constrained action mechanism is built.

### 3.4. Benefit of joint multi configuration optimization

Our approach of jointly optimizing a range of Polar code configurations represents a significant departure from prior learning based methods and is primarily motivated by the need to achieve learning efficiency at scale. Since the task of universal sequence design can be viewed as learning the

reliability ranking across synthetic channels, knowledge learned from smaller $K$ configurations can be transferred to accelerate convergence for larger $K$ cases. This leads to substantially higher learning efficiency while maintaining near optimal result.

The approach used in (Liao et al., 2023) limits training to a fixed $(N, K)$ code and relies on the agent's ability to generalize to other code configurations. Since the agent has not seen the decoder behavior for shorter payload lengths, its performance degrades significantly in this regime. Even when tasked to generate frozen bitmaps for larger payload sizes, which are presented in training, the agent's generalization performance remains suboptimal, and additional fine-tuning is required to close the optimality gap. Consequently, these weaknesses hinder practical deployment of this approach for online bitmap generation.

*Remark* 3.2. It is a general observation that in physical layer communications applications, the high precision required often makes generalization to unseen scenarios suboptimal unless strong domain structure is explicitly encoded as inductive bias. A practical rule of thumb is to include the entire test distribution during training even when reward generation induces additional computational burden. Reward-generation efficiency is further discussed in Appendix C.

*Remark* 3.3. An alternative to full test distribution coverage is to encourage fast adaptation via the meta learning framework (Finn et al., 2017). This represents a promising direction for future investigation.

### 3.5. Restrict search with limited lookahead

For short block lengths, it is possible to evaluate the rewards for all $K$'s in the fixed length-$N$ family of Polar codes. However, as we move to larger block lengths, joint optimization of all $K$'s is no longer effective. This is largely due to the use of a summarizing scalar reward, like averaging over the BLER of all Polar codes. The current action selection attempts to differentiate between minute values; the ability to find the true optimal value is completely swamped by a large number of noisy estimates.

To mitigate this smoothing effect, we rely on the empirical

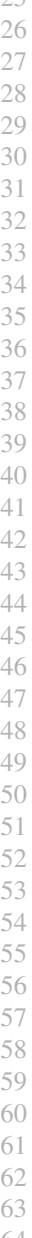

Performance delta @ 0.01 BLER - REF PW (beta expansion)

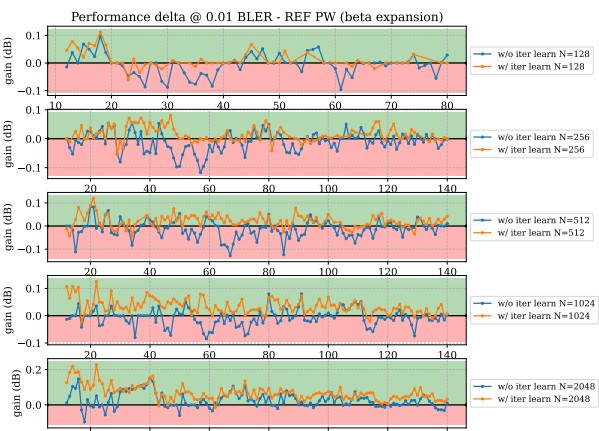

*Figure 2.* Performance impact of iterative learning in joint multi-configuration optimization.

---

**Algorithm 1** Neural-assisted sequence learning

> **function** SUBSEQSEARCH($\mathbf{s}$, $\mathbf{s}_l$, $I$, $W$)
>    ▷ Run UPO$^+$-GNN-PPO
>    $\pi \leftarrow \arg\max_\pi v_\pi(\mathbf{s}; \mathbf{s}_l, I + W)$    ▷ See (1)
>    $\mathbf{s} \leftarrow \text{greedy}(\pi | \mathbf{s})$
>    **return** $\mathbf{s}$
> **end function**

---

observation that the relative ordering of the indices will have little long range influence. In other words, the $J$-th decision will have only minor impact on the performance of a $(N, K)$ code where $K \gg J$. Using this, we modify the learning procedure such that for each of the agent's decisions, we perform rollouts up to a fixed lookahead window length $W$. Once we have committed to these decisions, we will learn the next decision by sliding the lookahead window forward. This is reminiscent of a game playing agent where it has a limited budget to search for an advantageous move, and once it executes the move, it will then proceed to evaluate the next move based on the current board configuration. We call this approach *iterative learning*.

In Figure 2 we compare the performance of joint multi-configuration optimization with iterative learning using a lookahead window of 16 to an earlier approach where we attempt to learn 512 configurations simultaneously. The learned sequence with iterative learning achieves gains across almost all configurations while we see degradation in a significant number of configurations without iterative learning.

In the next section, we will explain in detail the deep reinforcement learning techniques adopted in this work.

---

**Algorithm 2** Nested iterative sequence construction

**Require:** $N_{min}, N_{max}, K_{min}, W$
  Initialize $N \leftarrow N_{min}$, $\mathbf{s}_{lower} \leftarrow \emptyset$,
  **while** $N \leq N_{max}$ **do**
    ▷ Conduct search for a candidate sequence at $N$
    Initialize $K \leftarrow K_{min}$, ranking $\mathbf{r} \leftarrow \emptyset$
    **while** $K \leq N$ **do**
      **if** $K = K_{min}$ **then**
        ▷ Search for the first $K$ indices
        $\mathbf{s} = \text{SUBSEQSEARCH}(\emptyset, \mathbf{s}_{lower}, K, W)$
        $\mathbf{r} \leftarrow \mathbf{r} \cup \mathbf{s}_{1:K}$
      **else**
        ▷ Search for the $K$-th index
        $\mathbf{s} = \text{SUBSEQSEARCH}(\mathbf{r}, \mathbf{s}_{lower}, 1, W)$
        $\mathbf{r} \leftarrow \mathbf{r} \cup \mathbf{s}_K$
      **end if**
      $K \leftarrow K + 1$
    **end while**
    $\mathbf{s}_{lower} \leftarrow \mathbf{r}$
    $N \leftarrow 2N$
  **end while**
  **return** $\mathbf{r}$

---

## 4. Deep reinforcement learning

In our RL formulation, the agent's objective is to rank bit indices from the most to the least reliable, selecting one index at each decision step. Each RL task operates on a fixed code length $N$. During each episode, the agent is provided with a payload size $K$ and must output the $K$ bit indices used for carrying information bits in the $(N, K)$ code. The agent is rewarded based on a balance between the performance of the current code and that of all codes with larger payloads $(K' > K)$. The reward is issued only at the end of each episode, which corresponds to the negative BLER of the resulting code. The objective is thus to solve the following value-function maximization problem:

$$\pi_* = \arg\max_\pi v_\pi(\mathbf{s}; \mathbf{s}_l, W), \text{ for all } \mathbf{s} \in \mathcal{S}, \quad (1)$$

$$v_\pi(\mathbf{s}; \mathbf{s}_l, W) = \mathbb{E}_{\pi(\mathbf{s}, \mathbf{s}_l), K}[-\log(\text{BLER}_{K,N})],$$
$$K \sim \text{Uniform}(K(\mathbf{s}) : K_{max}), \quad (2)$$

where

1. $\mathbf{s}$ is the current state, $\mathbf{s} = (i_1, i_2, \ldots, i_{K-1})$ for some $K$.
2. $K(\mathbf{s})$ is the $K$ as defined in 1.
3. $K_{max} = \min(N, K + W)$.
4. $W$ is the lookahead window length.
5. $\pi(\mathbf{s}, \mathbf{s}_l)$ is the stochastic policy, with frozen past actions $\mathbf{s}$ and constrained under the UPO+ rule with embedded lower-N sequence $\mathbf{s}_l$.

6. BLER$_{K,N}$ is the block error rate of a $(N, K)$ Polar code with information bits mapped to the current $\pi(\mathbf{s}, \mathbf{s}_l)$ realization of bit indices.

### 4.1. States

The observed state can be viewed as an injective mapping of the agent's prior actions. We encode the state as a graph with nodes representing all synthetic bit channels, and setting the node type to selected or free $\{\mathbf{S}, \mathbf{F}\}$ based on the agent's actions. This ensures the Markovian property of the MDP problem is met. However, to reduce the message passing complexity over the graph, we confined the connections of nodes to a context window of past actions and the current action set. We have experimented with different context window size, i.e. the number of past actions to be considered in the policy network, and found that restricting the window to a small number of previous decisions was sufficient to maintain good performance. This truncation is reasonable given the observation of diminishing long-term influence of past decisions (Section 3.5).

### 4.2. Actions

As detailed in Section 3.2, the set of permissible actions at any given state $s$ is determined by the UPO+ rule – a composite constraint that combines the UPO relations, lower-N sequence enforcement, initial $K_{min}$ index relaxation, and node promotion. The relationship between the state and its permissible actions is encoded in the state-action transition function $A(s_{past}, s_{lower})$, where $s_{past}$ denotes the sequence of all past actions and $s_{lower}$ is the lower-N sequence. This logic is precisely captured in the UPO class of the accompanying source code.

### 4.3. Reward

The reward is defined as the negative log BLER of the $(N, K)$-Polar code produced by the policy at the end of each episode. The specific SNR for which the BLER is evaluated has a substantial impact on the value prediction of the actor-critic model. We anchor the SNR to the required SNR needed to achieve the target $1\%$ BLER for the reference beta-expansion sequence. This choice ensures that the evaluation point remains close to the operating region of the optimal sequence, as the observed performance gap between beta-expansion and other SOTA sequences is less than $0.2$ dB. We observe vastly improved value loss in the training compared to an alternate scheme using a constant capacity gap. This is defined as:

$$\text{SNR}_{\text{eval},K,N} = 10 \log_{10}(2^R - 1) + \text{SNR}_{\text{offset},N}, \quad (3)$$

where $R = K/N$ is the code rate and SNR is in dB. In Fig-

ure 3, the capacity gap, defined by the separation between the theoretical achievable SNR of a code and the SNR obtained from beta expansion at 1% BLER, is plotted over all configurations. It reveals that the gap is not constant across all configurations. Thus a fixed capacity gap increases variance in value prediction at convergence and creates a reward bias toward better performing code configurations.

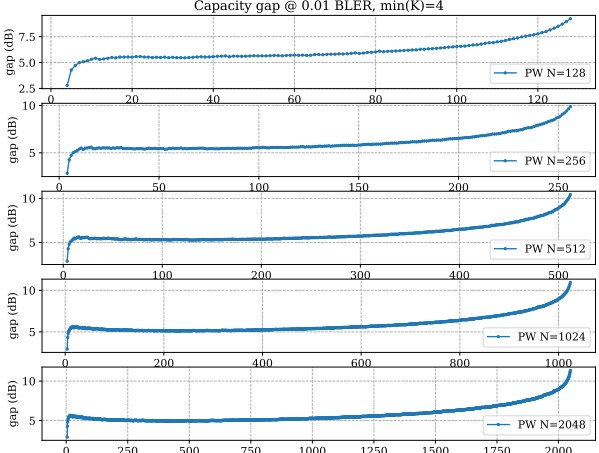

*Figure 3.* Required SNR characterization for baseline beta-expansion (PW) sequence used to define evaluation anchors.

The pseudo code of the iterative sequence construction method is given in Algorithms 1 and 2.

### 4.4. RL algorithm

After extensive experimentation with DQN including many of its enhancements (Mnih et al., 2015; Hessel et al., 2018) and PPO (Schulman et al., 2017), we adopted PPO as the primary algorithm for all universal sequence learning tasks. We observed stability issues with DQN early on (at $N = 256$), a substantial amount of effort was required to stabilize training. Furthermore, DQN's reliance on an experience replay buffer introduces synchronization overhead when parallelizing across GPU processes. This substantially limits throughput and scalability.

In contrast, we found the stable-baseline3 implementation of PPO (Raffin et al., 2021) to be highly stable and robust across all tested configurations, requiring minimal hyperparameter tuning. Extensive sweeps over learning rate, clipping ratio, GAE lambda and entropy regularization confirmed that PPO's performance remains largely insensitive to variations in these parameters. Additionally, PPO does not depend on a persistent replay buffer, it enables efficient data parallelism across compute nodes. This property proved critical for making N2048 training computationally feasible.

### 4.5. Actor-Critic Model

The architecture of the actor-critic model is largely determined by the chosen state representation. In our design, we elected to use a graph to encode the context and the action space, where each node in the graph maps to a bit index in the context or a bit index in the permissible action space. A graph representation provides a general description of the relations between nodes and additional semantic relations can be incorporated *a priori* or learned through gating or attentional mechanisms.

First we construct a graph of all synthetic channels each represented as a node in the graph. However, not all nodes are connected. Only the current context and the action space are fully connected to each other. Nevertheless, all nodes participate in the model's final decision.

Each node is assigned two input features. The first feature is a node type indicating whether the node has been selected (information carrying) or remains unselected (frozen). The second feature encodes the binary representation of the index itself, motivated by the observation that many theory-based selection schemes such as Reed-Muller and beta expansion are functions of the binary representation. An initial layer learns a type specific embedding and an embedding derived from the binary representation, which are combined using summation rather than concatenation – a choice that consistently yielded superior performance across trials.

The resulting embedding is passed to a stack of graph convolutional layers employing the GraphSAGE operator (Hamilton et al., 2017). We chose this operator because of its effectiveness and computational efficiency. We have experimented with many advanced gated (Bresson & Laurent, 2018) and attentional GNN architectures (Brody et al., 2022; Vaswani et al., 2017) but did not see appreciable performance gain from using them. We introduced techniques borrowed from Transformer architectures such as pre-layer normalization (pre-LN) and residual connections (Xiong et al., 2020). Both mechanisms show an appreciable impact on performance. ReLU activations are applied after each convolutional layer and prior to skip connection combining.

The model is multi-headed, sharing a common feature extraction body. The value head is implemented as an MLP operating on globally pooled node features, while the action head is a shared MLP that processes the node features in the current action set independently.

*Remark* 4.1. We observed that increasing network depth generally degraded performance, even when attentional mechanisms were present. This phenomenon suggests unresolved information propagation bottlenecks within GNN architectures, representing a promising topic for future investigation.

*Remark* 4.2. A key practical challenge in extending GNN-based designs to other Polar code variants lies in the absence of clearly established reliability relations among their synthetic channels. Learning such relations directly from decoder behavior constitutes an important avenue for future research.

### 4.6. Model Training

Training is conducted in a heterogeneous multi-node Kubernetes cluster. Each training instance uses a single Nvidia A100-40GB GPU node for GNN training and rollout orchestration. The reward generation for each parallelized environment is handled by a 8-core CPU worker node running a multi-threaded C++ based genie-aided (GA) SCL8 Polar decoder simulator. The GA decoder removes dependency on a particular CRC polynomial, which is critical because the standard employs different CRC polynomials for uplink and downlink. SCL8 is a successive cancellation decoder implementation with a list size of 8.

The BLER simulation employs a maximum of $500,000$ transmitted blocks and terminates early after 500 block errors to ensure high precision evaluation. Training continues until completion of the entire reliability sequence, independent of the total number of timesteps.

The Adam optimizer is used (Kingma & Ba, 2017) with a fixed learning rate of $2.5 \times 10^{-4}$. PPO hyper parameters are as follows: 32 rollout steps per update, batch size of 256, 10 epochs, clip range 0.1, discount factor $\gamma = 1.0$, and a lookahead window of 16. Sixteen parallel environments are executed concurrently. Specific to the GNN architecture, three convolutional layers are used, the initial embedding size and hidden features size are 64 for all convolutional layers. Both the value head and per node action head use an MLP with 2 hidden layers with 128 and 32 units respectively.

To maximize hardware utilization, the environment and rollout routines were modified to support asynchronous reward generation. Because reward signals are available only at episode termination and episode lengths are uniformly random, the default synchronous per step reward generation leads to low utilization, as only a subset of processes complete episodes at any given step. Instead, we implement a "lazy" rollout strategy that queues reward-generation requests during rollouts and processes them collectively at the end of each horizon, significantly increasing the throughput by maximizing utilization of the available CPU resources.

Training proceeds iteratively: initially $1,000$ episodes are used to learn the first 16 bit channels selections. Learning continues until the greedy policy consistently converges to the 16 selections (irrespective of order) across three consecutive evaluations. Thereafter the model learns subsequent

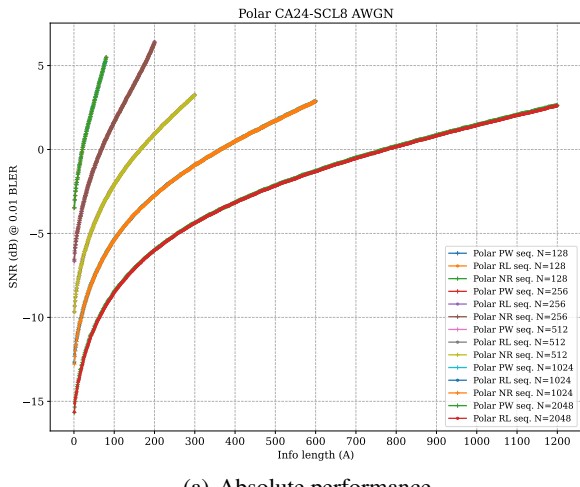

(a) Absolute performance

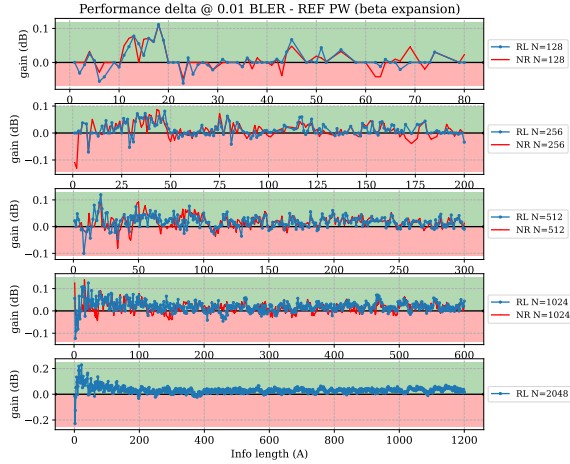

(b) Relative performance using PW as reference.

*Figure 4.* Polar code performance under CRC-aided successive cancellation list decoding over an AWGN channel for the beta expansion (PW), NR and RL-learned reliability sequences.

indices in pairs, requiring convergence to the same ordered selection for three consecutive evaluations before proceeding. This iterative procedure repeats until the full sequence is generated.

The majority of the overall training time is concentrated in the final progression (from N1024 to N2048), which required more than 30 days on the described cluster.

### 4.7. Pretraining

Although pretraining is not critical to the quality of the learned sequence, we nonetheless used a pretrained GNN for the final learning run to reduce the time required for settling to a stable solution. We used the GNN trained using our initial training method. This method is detailed in Appendix B.

## 5. Simulation Results

All sequences considered are evaluated using the NR CRC24C polynomial (3GPP, 2020), with BPSK modulation over an AWGN channel. CRC-aided SCL decoding with a list size of 8 is used throughout all experiments. The code blocks are constructed directly from the Polar transform without rate matching. The BLER vs SNR performance is evaluated across the waterfall region in 0.5dB increments. At each SNR point a maximum of $500,000$ blocks and a maximum limit of $500$ block errors is simulated. The required SNR to achieve $1\%$ BLER is extracted for each configuration and compared against the baseline beta expansion (PW) and the NR reliability sequence. Evaluations cover all

supported code rates up to approximately $CR = 0.6$.

In Figure 4(b), results are presented relative to the beta-expansion (PW) baseline. The relative performance of the NR sequence is also plotted for comparison. The RL-based method matches the performance of the NR sequence and, in particular, delivers consistent gains over the beta-expansion sequence at $N = 2048$. Both the learned sequence and NR sequence outperform the beta-expansion baseline with high probability. The learned sequence is included in Appendix A.

## 6. Conclusion

We present the first demonstration of a reinforcement-learning based universal sequence design at industrial scale, thereby making the approach practical for Polar code design in future communication system standardization efforts. We highlighted the critical use of the universal partial order – a law governing the relative reliability of the polarization induced synthetic channels – in constraining the search space and improving training efficiency. We also emphasized the advantage of joint multi-configuration optimization, which enables knowledge transfer across multiple code configurations and enhances learning efficiency.

Future work will explore extending this methodology to other Polar code variants by directly learning their intrinsic reliability structures and decoding specific properties, further advancing the data-driven design of channel coding techniques.

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

## A. Learned sequence

The following are side by side comparison of the RL-learned sequence, the beta expansion (PW) sequence and the NR sequence.

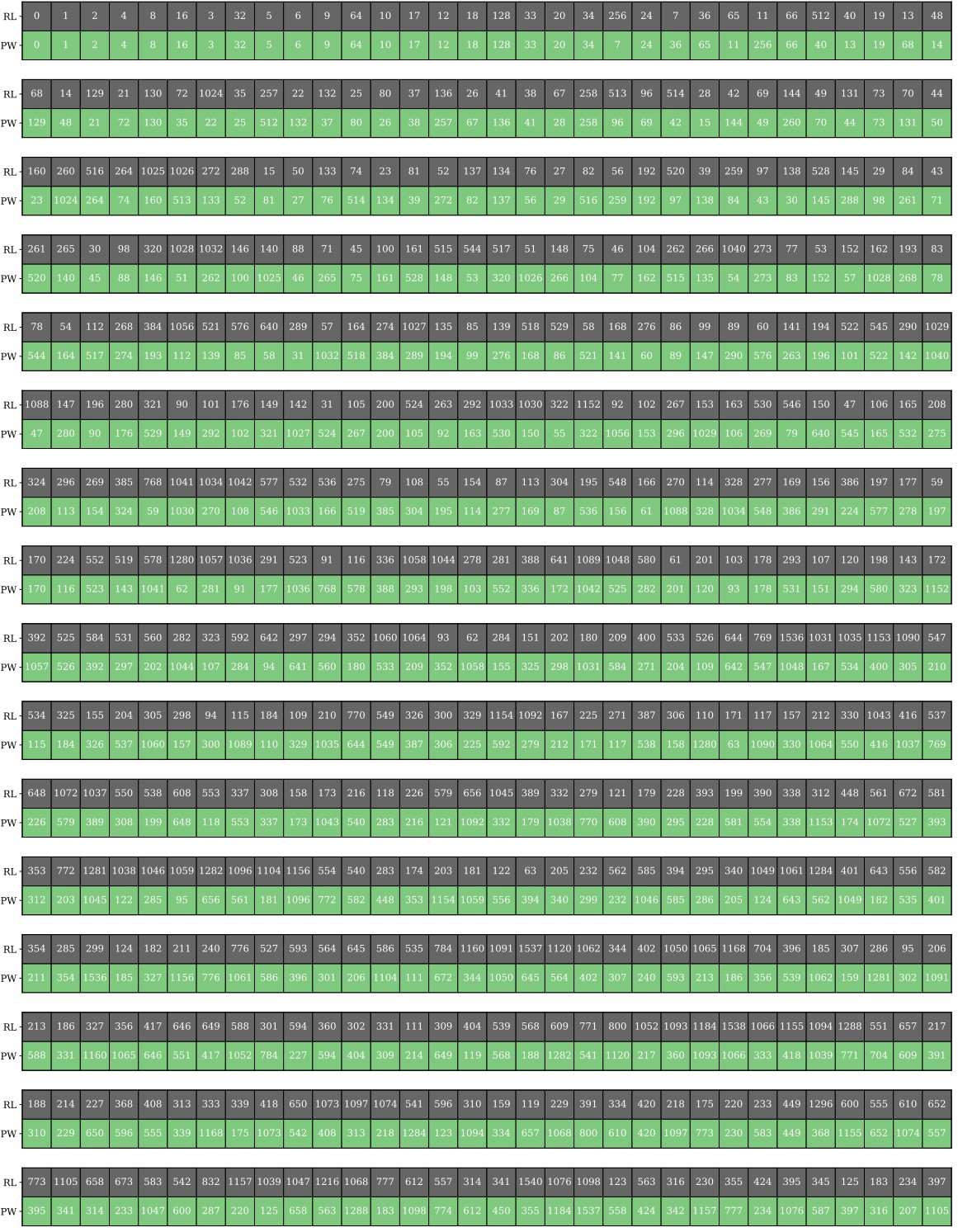

*Figure 5.* Comparison of RL-learned sequence vs beta expansion at $N = 2048$, first part.

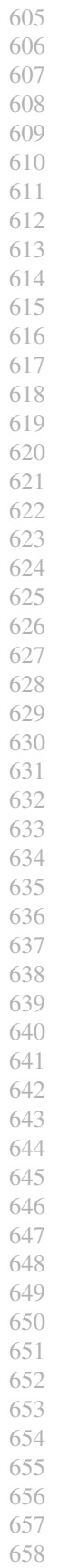

| RL | 432 | 450 | 587 | 660 | 1106 | 1158 | 1080 | 1312 | 241 | 674 | 1161 | 616 | 774 | 342 | 403 | 785 | 778 | 565 | 1051 | 1544 | 1063 | 357 | 398 | 207 | 187 | 236 | 346 | 215 | 242 | 287 | 452 | 558 |
| PW | 126 | 673 | 345 | 1051 | 660 | 565 | 403 | 241 | 1100 | 832 | 1538 | 452 | 187 | 357 | 1158 | 778 | 1063 | 616 | 398 | 1296 | 303 | 236 | 1106 | 589 | 674 | 346 | 1161 | 1080 | 647 | 566 | 432 | 1053 |

| RL | 1162 | 1121 | 1283 | 1122 | 664 | 1100 | 705 | 624 | 786 | 896 | 358 | 361 | 405 | 348 | 126 | 456 | 569 | 595 | 589 | 566 | 676 | 1552 | 1108 | 1169 | 1170 | 1285 | 780 | 647 | 1344 | 706 | 801 | 419 |
| PW | 785 | 242 | 595 | 405 | 215 | 664 | 358 | 569 | 1540 | 189 | 1283 | 1216 | 780 | 1121 | 590 | 456 | 361 | 1162 | 1108 | 1067 | 676 | 348 | 419 | 1054 | 786 | 705 | 624 | 406 | 311 | 244 | 651 | 597 |

| RL | 1053 | 680 | 590 | 189 | 219 | 244 | 303 | 406 | 1185 | 369 | 409 | 248 | 362 | 311 | 464 | 1164 | 1286 | 1067 | 1095 | 1112 | 1075 | 1124 | 1054 | 1539 | 570 | 651 | 597 | 611 | 802 | 788 | 421 | 601 |
| PW | 570 | 1169 | 190 | 1312 | 543 | 409 | 1122 | 219 | 1285 | 362 | 1095 | 1544 | 335 | 1164 | 896 | 1069 | 801 | 706 | 611 | 421 | 1112 | 788 | 231 | 680 | 598 | 464 | 369 | 1170 | 653 | 1075 | 572 | 410 |

| RL | 708 | 1289 | 1568 | 1069 | 1297 | 653 | 572 | 688 | 792 | 231 | 335 | 364 | 190 | 598 | 1128 | 1290 | 659 | 425 | 221 | 422 | 370 | 451 | 372 | 410 | 613 | 235 | 343 | 315 | 317 | 480 | 712 | 833 |
| PW | 315 | 248 | 1286 | 601 | 1124 | 221 | 364 | 659 | 1070 | 802 | 1289 | 422 | 1099 | 775 | 708 | 613 | 451 | 370 | 1185 | 654 | 1552 | 559 | 425 | 343 | 1172 | 792 | 235 | 1077 | 602 | 412 | 317 | 222 |

| RL | 1186 | 1172 | 543 | 804 | 1408 | 1077 | 1099 | 1313 | 1070 | 412 | 654 | 661 | 602 | 453 | 775 | 1292 | 1298 | 1136 | 1541 | 1188 | 1078 | 1314 | 1107 | 614 | 617 | 376 | 222 | 834 | 1600 | 1542 | 604 | 237 |
| PW | 1344 | 127 | 688 | 1290 | 661 | 1128 | 804 | 614 | 480 | 1101 | 833 | 1186 | 1539 | 453 | 426 | 372 | 1159 | 779 | 712 | 1078 | 617 | 399 | 1297 | 318 | 237 | 1107 | 604 | 675 | 347 | 1176 | 1081 | 662 |

| RL | 454 | 433 | 426 | 559 | 675 | 808 | 1081 | 1101 | 1217 | 1176 | 1159 | 1123 | 1082 | 1218 | 720 | 665 | 662 | 779 | 618 | 677 | 625 | 359 | 1192 | 318 | 347 | 243 | 399 | 127 | 238 | 434 | 457 | 428 |
| PW | 567 | 433 | 1292 | 243 | 1102 | 834 | 1568 | 454 | 665 | 359 | 1188 | 808 | 1541 | 618 | 428 | 1298 | 1217 | 781 | 238 | 1136 | 591 | 457 | 376 | 1163 | 1109 | 1082 | 677 | 349 | 434 | 1055 | 787 | 720 |

| RL | 836 | 816 | 1102 | 897 | 567 | 1163 | 1545 | 1546 | 787 | 245 | 781 | 736 | 591 | 626 | 1109 | 1110 | 666 | 458 | 620 | 707 | 349 | 782 | 571 | 436 | 363 | 1287 | 1300 | 407 | 350 | 465 | 789 | 681 |
| PW | 625 | 407 | 245 | 666 | 836 | 571 | 1542 | 1408 | 191 | 1313 | 1218 | 782 | 1123 | 620 | 458 | 1300 | 363 | 1192 | 1110 | 1545 | 678 | 350 | 1165 | 897 | 1084 | 816 | 707 | 626 | 436 | 1113 | 789 | 246 |

| RL | 840 | 678 | 898 | 599 | 668 | 803 | 1345 | 1346 | 1316 | 1220 | 1125 | 1553 | 1554 | 1055 | 1548 | 1171 | 1664 | 1165 | 1166 | 1113 | 1114 | 1304 | 628 | 460 | 246 | 573 | 709 | 249 | 423 | 1084 | 371 | 1200 |
| PW | 681 | 599 | 465 | 1171 | 668 | 1314 | 573 | 411 | 1600 | 249 | 1287 | 1220 | 840 | 1125 | 1546 | 460 | 365 | 1166 | 898 | 1071 | 803 | 736 | 1304 | 423 | 1114 | 790 | 709 | 682 | 628 | 466 | 371 | 1200 |

| RL | 440 | 466 | 682 | 1224 | 689 | 411 | 365 | 848 | 250 | 413 | 366 | 191 | 373 | 468 | 481 | 805 | 793 | 900 | 790 | 1173 | 1320 | 1116 | 1556 | 1348 | 1126 | 1174 | 1129 | 1232 | 1177 | 1569 | 1187 | 1409 |
| PW | 655 | 1553 | 574 | 440 | 1173 | 793 | 250 | 1316 | 603 | 413 | 1126 | 223 | 1345 | 366 | 689 | 1548 | 1291 | 1224 | 900 | 1129 | 805 | 710 | 615 | 481 | 1116 | 848 | 1187 | 684 | 1554 | 468 | 427 | 373 |

| RL | 632 | 603 | 713 | 615 | 710 | 690 | 684 | 574 | 655 | 835 | 414 | 427 | 223 | 374 | 252 | 482 | 794 | 809 | 605 | 619 | 721 | 904 | 1291 | 1328 | 1189 | 1071 | 1293 | 1137 | 806 | 864 | 1570 | 1560 |
| PW | 1174 | 794 | 713 | 1079 | 632 | 414 | 319 | 252 | 1346 | 605 | 690 | 1177 | 1320 | 663 | 1130 | 806 | 1293 | 482 | 1103 | 835 | 1664 | 1569 | 455 | 904 | 374 | 1189 | 809 | 714 | 1556 | 619 | 429 | 1299 |

| RL | 1130 | 1079 | 1248 | 1572 | 714 | 663 | 692 | 1132 | 1193 | 429 | 455 | 319 | 435 | 377 | 472 | 484 | 837 | 817 | 796 | 912 | 1601 | 1178 | 1352 | 1299 | 1792 | 1219 | 1190 | 1103 | 1138 | 1410 | 1294 | 810 |
| PW | 1232 | 796 | 239 | 1137 | 606 | 472 | 377 | 1178 | 1348 | 1083 | 692 | 435 | 1294 | 721 | 1132 | 864 | 1570 | 484 | 667 | 837 | 1190 | 810 | 1543 | 1409 | 430 | 1328 | 1219 | 783 | 716 | 1138 | 621 | 459 |

| RL | 679 | 722 | 627 | 737 | 667 | 621 | 378 | 437 | 459 | 696 | 899 | 716 | 488 | 1543 | 1301 | 1602 | 1180 | 1083 | 1221 | 430 | 461 | 239 | 669 | 606 | 1412 | 1315 | 1140 | 818 | 841 | 838 | 1111 | 1201 |
| PW | 1301 | 378 | 1193 | 1111 | 1560 | 679 | 351 | 1180 | 912 | 1085 | 817 | 722 | 627 | 437 | 1352 | 247 | 696 | 838 | 1572 | 1410 | 669 | 1315 | 1248 | 812 | 1601 | 622 | 488 | 1302 | 1221 | 841 | 1194 | 1140 |

| RL | 629 | 928 | 812 | 738 | 622 | 724 | 1576 | 1360 | 1547 | 1194 | 1665 | 1085 | 1302 | 1305 | 820 | 849 | 380 | 351 | 467 | 1317 | 1604 | 1225 | 1202 | 1416 | 1222 | 1144 | 438 | 1196 | 441 | 469 | 247 | 630 |
| PW | 1547 | 461 | 380 | 1167 | 899 | 1086 | 818 | 737 | 1305 | 438 | 1115 | 791 | 724 | 683 | 629 | 467 | 1201 | 670 | 1792 | 575 | 441 | 1602 | 1412 | 251 | 1317 | 1222 | 842 | 1127 | 1576 | 462 | 1360 | 367 |

| RL | 415 | 462 | 367 | 496 | 740 | 728 | 901 | 783 | 1584 | 1167 | 791 | 711 | 683 | 1086 | 1376 | 1127 | 1318 | 1549 | 1204 | 691 | 670 | 902 | 842 | 1555 | 1306 | 1321 | 1115 | 850 | 685 | 633 | 960 | 844 |
| PW | 1196 | 928 | 1549 | 738 | 1306 | 1225 | 901 | 1144 | 820 | 711 | 630 | 496 | 1117 | 849 | 1202 | 685 | 1555 | 469 | 442 | 1175 | 795 | 728 | 1318 | 633 | 415 | 1604 | 253 | 1347 | 844 | 691 | 1550 | 1416 |

| RL | 905 | 442 | 1424 | 1226 | 1175 | 1308 | 1666 | 1550 | 1233 | 375 | 483 | 575 | 473 | 444 | 1117 | 1329 | 1608 | 1228 | 1347 | 470 | 251 | 807 | 865 | 715 | 795 | 686 | 824 | 1557 | 1349 | 1668 | 1131 | 1133 |
| PW | 1321 | 1226 | 902 | 1131 | 807 | 740 | 1308 | 483 | 1118 | 850 | 1665 | 686 | 1584 | 470 | 905 | 375 | 1204 | 824 | 715 | 1557 | 634 | 444 | 1233 | 797 | 254 | 1376 | 607 | 473 | 1179 | 1349 | 1322 | 693 |

| RL | 1249 | 1571 | 693 | 744 | 1616 | 1573 | 1353 | 634 | 852 | 636 | 797 | 906 | 811 | 717 | 485 | 379 | 431 | 489 | 474 | 253 | 1322 | 1191 | 1793 | 1118 | 1234 | 1208 | 1561 | 1139 | 1440 | 1324 | 1179 | 1350 |
| PW | 1608 | 1295 | 1228 | 960 | 1133 | 865 | 1666 | 1571 | 485 | 906 | 852 | 1191 | 811 | 744 | 1558 | 1424 | 431 | 1329 | 1234 | 798 | 717 | 1139 | 636 | 474 | 379 | 1208 | 1350 | 1561 | 694 | 1181 | 913 | 1324 |

*Figure 6.* Comparison of RL-learned sequence vs beta expansion $N = 2048$, second part.

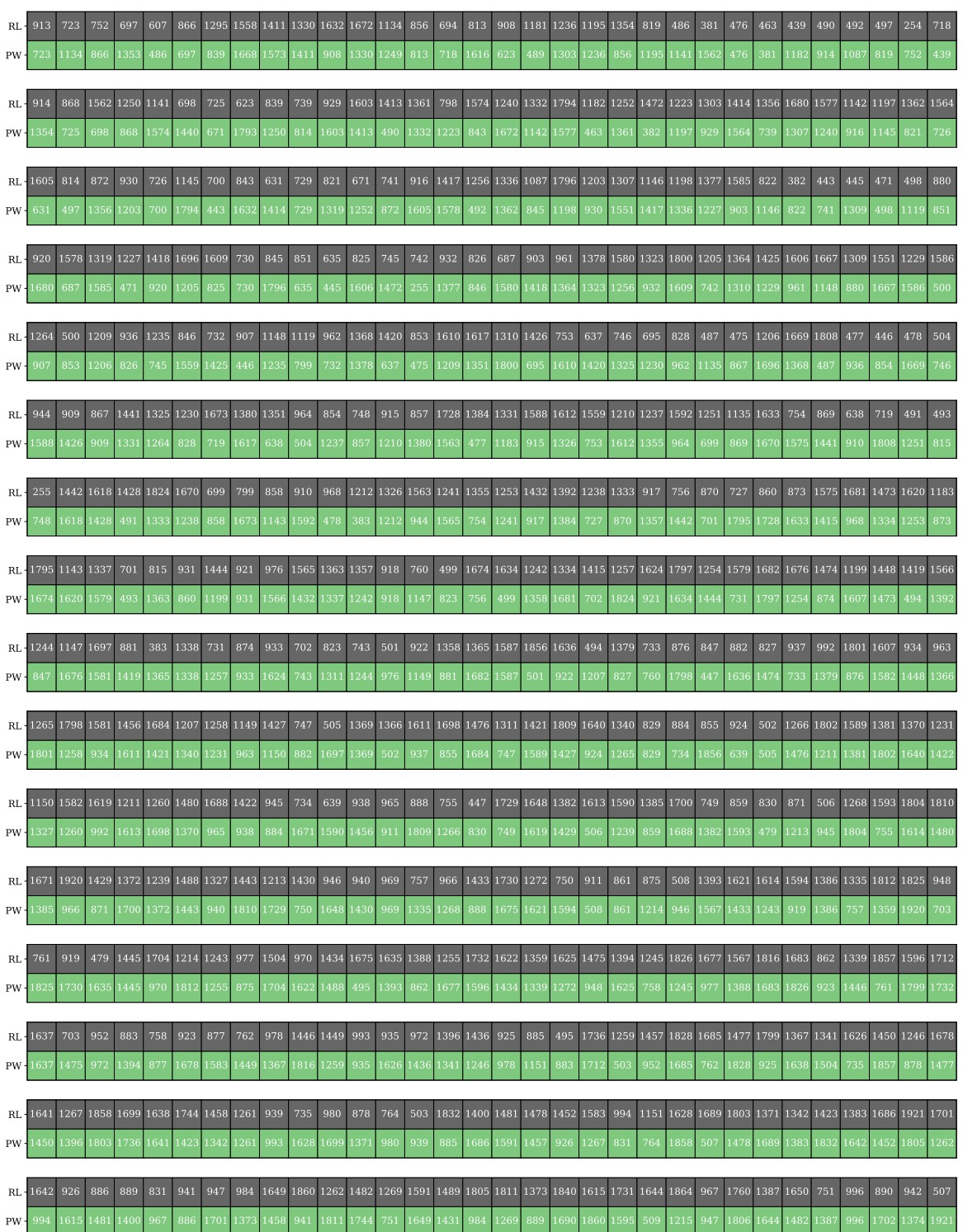

*Figure 7.* Comparison of RL-learned sequence vs beta expansion $N = 2048$, third part.

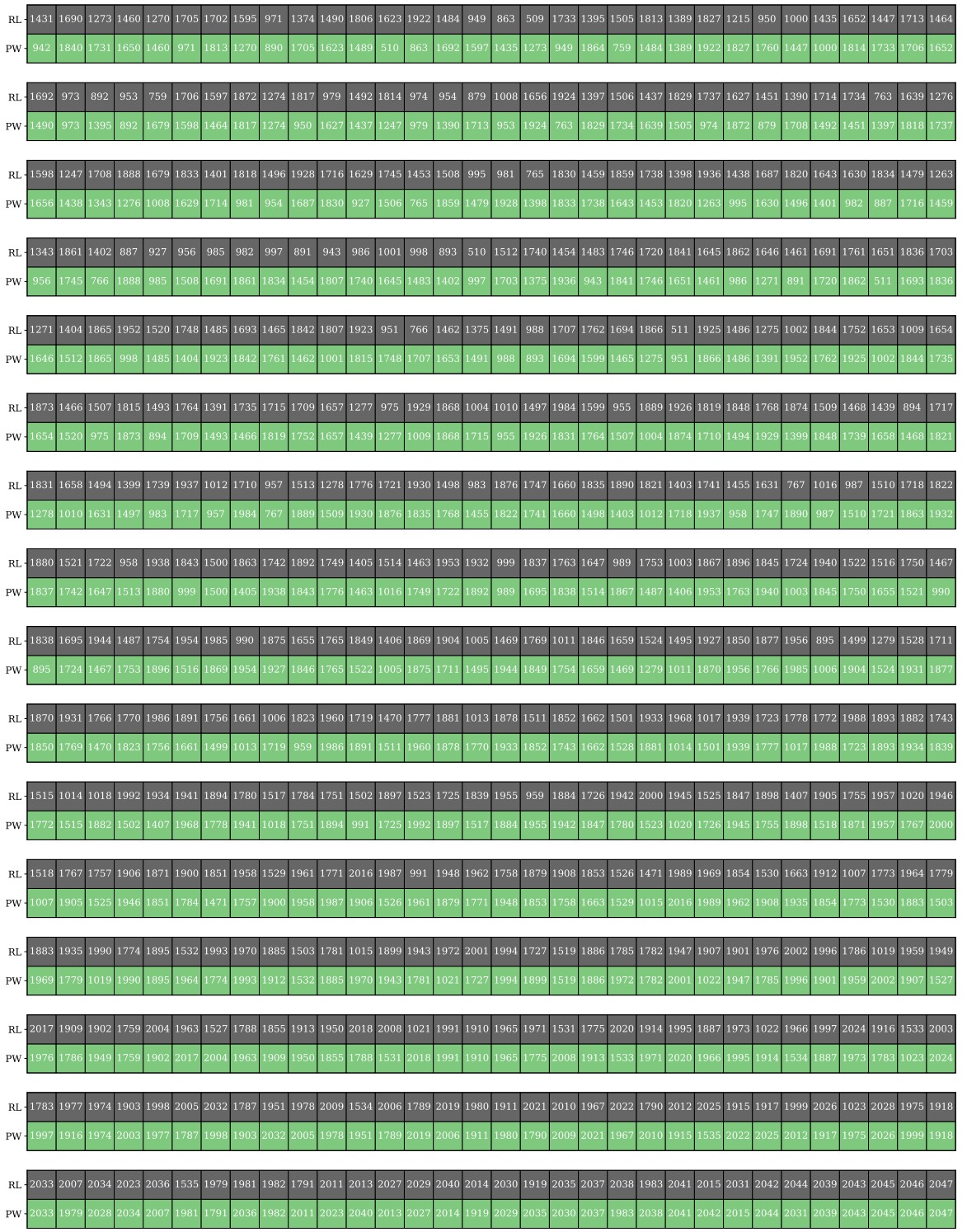

*Figure 8.* Comparison of RL-learned sequence vs beta expansion $N = 2048$, fourth part.

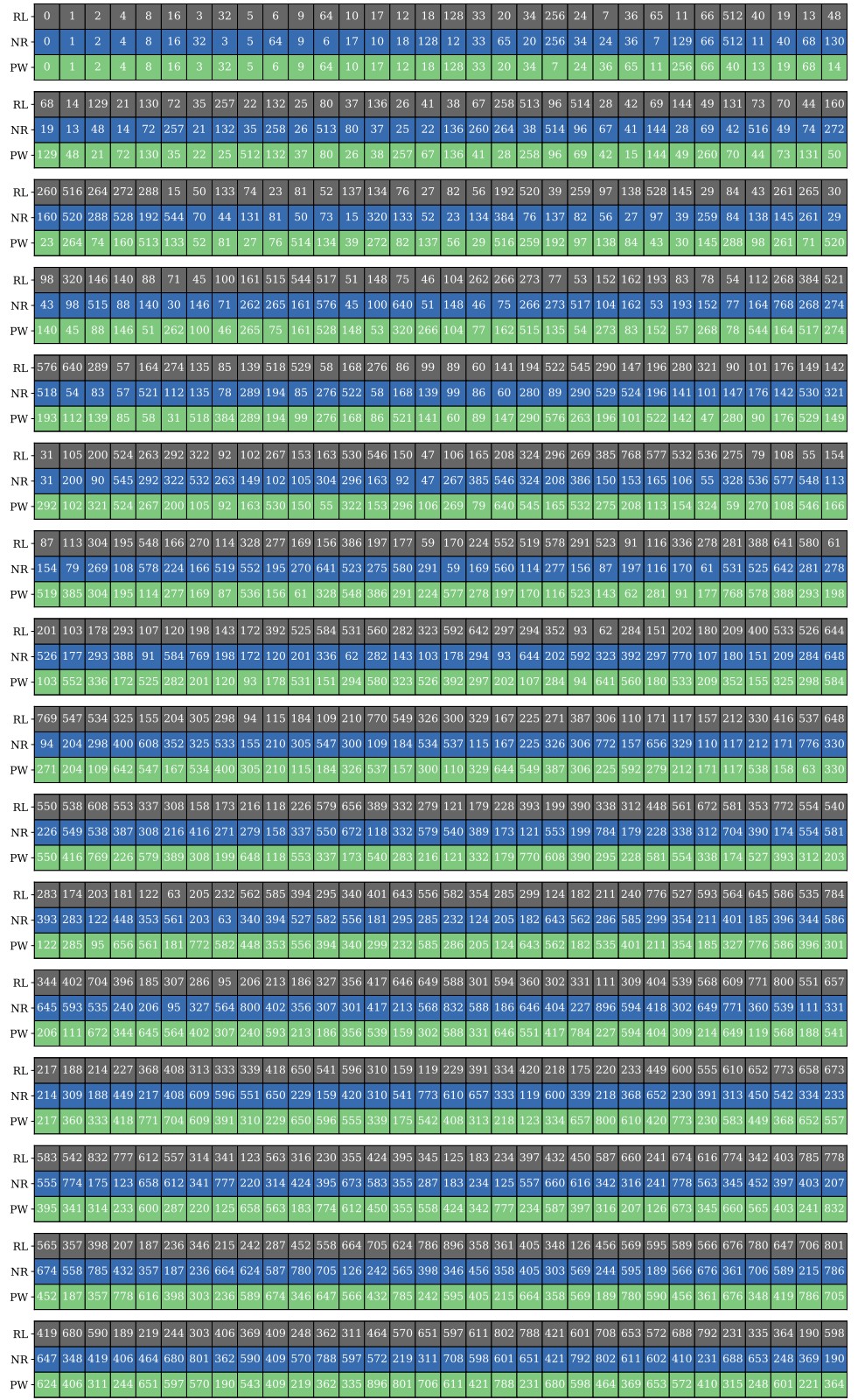

*Figure 9.* Comparison of RL-learned sequence, beta expansion sequence and NR sequence at $N = 1024$, first part.

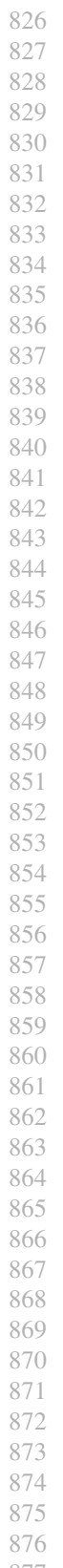

*Figure 10.* Comparison of RL-learned sequence, beta expansion sequence and NR sequence at $N = 1024$, second part.

## B. GNN Pretraining

In this section, we describe the initial sequence learning method which was used to pretrain the model.

### B.1. Whole sequence learning with range expansion

For any sequence length $N \leq 512$, all configurations are optimized jointly: the agent experiences the outcome of all $(N, K)$ codes in a single run. In this mode, we found that learning becomes much more efficient if we begin with a small range of configurations (e.g. 64) and gradually increase the range up to the maximum. Thus we employ a schedule that increase the number of configurations by 64 at regular intervals until it reaches the full range at about $50\%$ of training.

### B.2. Multi-stage training

At $N = 2048$, we split the sequence into 4 segments and train each segment individually, while holding all previously learned segments fixed. This was needed to reduced the smoothing effect of averaging many rewards over the range of configurations considered.

## C. Runtime efficiency

After incorporating UPO and reducing graph connections to local context and actions, the main runtime bottleneck was reward generation. In the reference method (Liao et al., 2023), only a single configuration is learned in a single RL-training run, it requires generating a reward (running a BLER simulation) at every step within an episode. For a $(128, 64)$ code, every episode takes 64 steps and around $10,000$ episodes are used to train the model. This is only one single configuration.

By moving to universal sequence learning, we train all configurations jointly and each episode needed a single BLER simulation at termination. This reduced the BLER simulations requirement by orders of magnitude. Second, by running a multi-threaded multi-core CPU C++ Polar simulator implementation we achieved a 80x speed up compared to a standard Matlab implementation. We ran 16 environments in parallel at the largest sequence length thus producing another 16x speed up in training.

As an illustration, at $N = 1024$, we achieve 6 orders of magnitude improvement in runtime over the reference method using serial environment execution and standard Matlab based simulator.

### C.1. GPU-based Polar simulator implementation

We also evaluated the execution time for a data-parallel GPU-based implementation in Pytorch, with the initial belief that exploiting the SIMD architecture will lead to substantial runtime improvement. The simulator is configured to dispatch 1000 data blocks in parallel. However, the GPU implementation did not significantly improve over a single-core CPU C++ based implementation. We suspect that the simulator, which includes the encoder, noise generation and decoder, was sufficiently small that instruction and data are kept in cache for the entire C++ simulation run. We did not consider JIT compilation of the Pytorch implementation since we only had mixed successes with the feature and we do not expect it will provide the 7x improvement needed to match a 8-core C++ implementation.

## D. Training pipeline implementation

We document all software components used to facilitate implementation. The training pipeline is implemented in PyTorch[3] using the PyTorch Geometric (PyG) [4] library for graph machine learning and the PPO implementation from stable-baseline3[5]. Multi-node orchestration utilizes a custom dask[6]-based framework. The multi-threaded Polar decoder simulator is written in C++ using the native threading library with python/C++ interoperability handled via pybind11[7].

---

[3]https://pytorch.org/
[4]https://pytorch-geometric.readthedocs.io/
[5]https://stable-baselines3.readthedocs.io/
[6]https://www.dask.org/
[7]https://pybind11.readthedocs.io/

