# OpenReview forum: "A Reinforcement Learning Based Universal Sequence Design for Polar Codes"
_ICML.cc/2026/Conference — Submitted to ICML 2026_

### Official Review · Reviewer_H4v7 · 2026-02-23

**Soundness:** 3
**Presentation:** 3
**Significance:** 4
**Originality:** 3
**Overall Recommendation:** 6
**Confidence:** 4

**Summary:**

This paper proposes an RL-based polar code construction algorithm that can generate a universal sequence that can work across all block / bit lengths, instead of creating bespoke codes for each block and bit length. The authors utilize physical constraints (UPO) to narrow down the search space, and also use scheduled curriculum learning that utilizes results from shorter block lengths to construct codes for longer block lengths. The authors also propose two annealing strategies to ensure good performance at longer block lengths.

**Compliance With Llm Reviewing Policy:**

Affirmed.

**Final Justification:**

There is no score over 6. My score cannot change.

**Key Questions For Authors:**

Q1: The main advantage of this RL-aided polar code design scheme would be the reduction of code design time and the reduction of memory / compute for UEs that work under extended block lengths. Can the authors provide a numerical result that outlines how much savings the RL-based methodology can provide? While the manual design scheme would be hard to quantify in terms of design time, an industry standard time (e.g. man-months) would suffice for this comparison.

Q2: While the current method works up to a block length of 2048, 6G communications may require block lengths that are longer than this, due to extremely stringent requirements on the control channels. However, by looking at the abstract, it seems that N2048 was the limit of the current work. What should be done to adapt this work to block lengths of say, 4096 or 8192?

Q3: The restricted search with limited lookahead in section 3.5 seems to be a key factor in long-sequence design, but the overall flow of the algorithm is extremely hard to visualize. Can the authors rephrase this algorithm?

**Limitations:**

Yes

**Strengths And Weaknesses:**

Soundness: The manuscript is soundly written, with all lemmans and definitions clearly written. The rather simple information-theoretic approach to enforcing physical constriaints in reducing the search space is easily written enough to understand.

Presentation: The presentation is clear, with the algorithm and equations easy to understand. The key principles are clear.

Significance: This paper has high significance in the AI-RAN community. A universal design paradigm for polar codes exceeding block lengths of 1024 is a highly sought-out-for criteria, and this RL-based design paridigm is a game changer.

Originality: While the methodology is an evolutionary change (UPO is a well-known concept in coding theory), the combination of UPO as a physical prior to RL is original.

---

> ### Author Rebuttal · Authors · 2026-03-28
>
> Thank you for taking the time to review the paper.  Let me address your questions in turn.
>
>
> In regards to Q1, I would like to clarify that we are designing an extended nested universal sequence, so it supports all 5G NR lengths and code rates, in addition to N=2048.  The actual development of the RL-based algorithm took 8 months, however, the time it takes to create a new N2048 universal sequence can be achieved within 45 days.  I am the AIML-designer of the RL-based algorithm, with support from the second author who is a Polar Codes Expert.  Thus the task can be accomplished by one person with some channel coding consultation.
> The 5G NR sequence was designed by Huawei; they did not publicly release their design method.  However, it’s known from private conversations that they ran a heuristic design algorithm that was extremely resource intensive.  They implemented the Polar code simulator in FPGA and ran for an extended period of time.  The effort required a team of engineers to accomplish.  Thus, this work democratizes the polar codes design such that any institution who has access to cloud computing resources can design a sequence within a reasonable time budget.  This opens the door for custom design targeting unique, uncommon channel environments and other perhaps more exotic decoder implementations.
>
> In regards to Q2, The current runtime bottleneck is the reward generation step, each reward is a BLER simulation of 500,000 code blocks.  It’s certainly possible to increase the parallelism of the simulation from 8x to 16x or 32x.  That should make 4096 and 8092 attainable.  However, in my opinion adapting the design for PAC codes and other stronger Polar code variants will further improve the performance of the Polar code family, this is our intended next step for RL-based code design.
>
> In regards to Q3, thank you for the comment.  I will address this in the next revision of the paper by adding a diagram for ease of visualization.  Due to the limit on the paper length for ICML, we might have to release this addition in arXiv.  Also, we will release the source code of the project which allows unambiguous dissemination of the algorithm.

---

> > ### Author Rebuttal · Reviewer_H4v7 · 2026-04-01
> >
> > Perfect, thank you for your answers. Since there is no score over 6, my strong accept recommendation still holds.

---

### Official Review · Reviewer_5zwb · 2026-03-12

**Soundness:** 3
**Presentation:** 3
**Significance:** 2
**Originality:** 2
**Overall Recommendation:** 3
**Confidence:** 4

**Summary:**

The paper addresses the problem of universal sequence design (the sequence determines frozen symbols for different rates and length). The problem is indeed important as now the choice is done mostly by simulations. The authors propose a method based on the reinforcement learning. The final results show the same performance as 5G polar codes for $N \leq 1024$. The authors also provided the results for $N=2048$ (no 5G case available).

**Compliance With Llm Reviewing Policy:**

Affirmed.

**Final Justification:**

The method is indeed interesting, but the method should be supported by resulting improvements. So in my opinion it is mandatory to present the codes constructed by this method, which outperform the existing ones. In the current paper we see that the performance curves coincide (N < 1024) and for N=2048 there are no 5G codes (and comparison is done with some bad codes).

**Key Questions For Authors:**

- The discussion on the larger lengths is questionable. It is known that LDPC codes do outperform polar codes for $N > 1024$, this is the main reason on length constraint.
- The benefits of the new method is not evident. Please compare the complexities of the new method and the method used to construct 5G LDPC codes.

**Limitations:**

yes

**Strengths And Weaknesses:**

The method itself is interesting as it utilizes subchannels’ ordering and is faster than existing RL-based methods.

At the same time the main concern is performance. It is more less the same as for 5G polar codes. So there is no clear evidence why one should use the proposed method. The discussion on the larger lengths is questionable. It is known that LDPC codes do outperform polar codes for $N > 1024$, this is the main reason on length constraint.

---

> ### Author Rebuttal · Authors · 2026-03-28
>
> Thank you for taking the time to review the paper.  Let me address your questions in turn.
>
>
> In regards to Q1, you are correct in pointing out that at the time channel codes investigations were conducted for 5G (2016), LDPC codes did outperform Polar Codes at long codeword lengths.  However, major developments have taken place since then, in particular, with the introduction of polarization-adjusted convolutional (PAC) codes, the extended Polar codes family can reach the dispersion bound, thus research in the Polar codes family remains very active.  PAC codes improve performance by adding a simple rate-1 convolutional transform in the precoding stage, and the same low complexity decoding strategy can be used to decode them, making them very attractive for practical deployment.
>
> Second, as we have stated in the paper, Polar codes remain the encoding strategy for control channel (PDCCH) for 6G, and all industry participants are actively working on enhancing the performance of Polar codes for this iteration.  The additional control signaling demand and the continued need to improve cell edge experience mean increasing code length is highly desirable.  Thus there is no constraint to the length of Polar codes; the performance deficit can be overcome with new techniques.
>
> In regards to Q2, let me clarify that the goal of the paper is to address the scalability of current RL-based Polar codes design methods.  The complexity of these methods restricted their use to short code lengths, i.e. N=256.
>
> The main contributions of the paper are the use of physical law constrained learning by exploiting the Universal Partial Order (UPO) property of Polar Codes to constrain the search space, the exploitation of weak long term influence of decisions to limit lookahead evaluation and the joint multi-configuration optimization to promote knowledge transfer among configurations, leading to substantially improved learning efficiency.  We accomplished these efficiency gains, while still maintaining the nested property of the universal sequence, through the use of lower-N sequence embedding.  The nested property is crucial for the high variability of the wireless environment because the derived codes can operate at near optimal performance for any code lengths N={128,256,512,1024,2048} and all payload lengths using this single sequence.  RL-based code designs to date limited their focus on designing a single configuration, i.e. one code rate, thus each configuration required a redesign.  I invite the reviewer to take a more careful examination of this work to fully appreciate the design constraints, the scale of the problem and the combination of myriad techniques to achieve these results.
>
> Thus, it is not the intention of this work to develop a Polar code variant that outperforms LDPC codes.  This work lays the foundation for RL-based polar code design methods at scale. The design of these Polar code variants using RL-based data driven methods, as stated in the conclusion section of the paper, will be the target of future work.

---

### Decision · Program_Chairs · 2026-04-30

**Decision:**

Reject

**Comment:**

This paper aims to advance Polar code design for 6G applications by developing a reinforcement learning–based universal sequence design framework. However, the proposed work remains limited to construction of codes that are comparable to prior work with significant questions remaining as how to achieve the original aim.

The paper is correct and the proposed approach is of interest and good amount of novelty.  However, the work seems to fall short in terms of impact and completeness. I agree with the second reviewer that the findings seem to be, at best, the very first step in designing Polar code at scale. As such, I also find the study rather incomplete and rushed for a full-fledge high impact publication. In fact, I find that consistent with the authors' conclusion, in the original manuscript, that the design of good Polar code variants -- those that can outperform LDPC codes and/or consistent w PAC construction-- using RL-based data driven methods, is an important area of future work.